# Arylphthalide Delays Diabetic Retinopathy via Immunomodulating the Early Inflammatory Response in an Animal Model of Type 1 Diabetes Mellitus

**DOI:** 10.3390/ijms25158440

**Published:** 2024-08-02

**Authors:** Francisco Martín-Loro, Fátima Cano-Cano, María J. Ortega, Belén Cuevas, Laura Gómez-Jaramillo, María del Carmen González-Montelongo, Jan Cedric Freisenhausen, Almudena Lara-Barea, Antonio Campos-Caro, Eva Zubía, Manuel Aguilar-Diosdado, Ana I. Arroba

**Affiliations:** 1Diabetes Mellitus Laboratory, Instituto de Investigación e Innovación en Ciencias Biomédicas de la Provincia de Cádiz (INiBICA), Hospital Universitario Puerta del Mar, 11009 Cádiz, Spain; francisco.martin@inibica.es (F.M.-L.); fatima.cano@inibica.es (F.C.-C.); belen.cuevas@inibica.es (B.C.); laura.gomez@inibica.es (L.G.-J.); mcarmen.gonzalez@inibica.es (M.d.C.G.-M.); manuel.aguilar.sspa@juntadeandalucia.es (M.A.-D.); 2Departamento de Química Orgánica, Facultad de Ciencias del Mar y Ambientales, Universidad de Cádiz, 11510 Puerto Real, Spain; mariajesus.ortega@uca.es (M.J.O.); eva.zubia@uca.es (E.Z.); 3Dermatology and Venereology Division, Department of Medicine, Karolinska Institute, SE-171 77 Solna, Sweden; cedric.freisenhausen@medsci.uu.se; 4Center for Molecular Medicine, Karolinska University Hospital, SE-171 76 Solna, Sweden; 5Department of Endocrinology and Metabolism, University Hospital Puerta del Mar, 11009 Cádiz, Spain; almlarbar@gmail.com; 6Área Genética, Departamento Biomedicina Biotecnología y Salud Pública, Universidad de Cádiz, 11510 Puerto Real, Spain; antonio.campos@uca.es

**Keywords:** diabetic retinopathy, type 1 diabetes mellitus, arylphthalides, microglia, inflammation, M2 response, immunomodulation, HO1

## Abstract

Diabetic retinopathy (DR) is one of the most prevalent secondary complications associated with diabetes. Specifically, Type 1 Diabetes Mellitus (T1D) has an immune component that may determine the evolution of DR by compromising the immune response of the retina, which is mediated by microglia. In the early stages of DR, the permeabilization of the blood–retinal barrier allows immune cells from the peripheral system to interact with the retinal immune system. The use of new bioactive molecules, such as 3-(2,4-dihydroxyphenyl)phthalide (M9), with powerful anti-inflammatory activity, might represent an advance in the treatment of diseases like DR by targeting the immune systems responsible for its onset and progression. Our research aimed to investigate the molecular mechanisms involved in the interaction of specific cells of the innate immune system during the progression of DR and the reduction in inflammatory processes contributing to the pathology. In vitro studies were conducted exposing Bv.2 microglial and Raw264.7 macrophage cells to proinflammatory stimuli for 24 h, in the presence or absence of M9. Ex vivo and in vivo approaches were performed in BB rats, an animal model for T1D. Retinal explants from BB rats were cultured with M9. Retinas from BB rats treated for 15 days with M9 via intraperitoneal injection were analyzed to determine survival, cellular signaling, and inflammatory markers using qPCR, Western blot, or immunofluorescence approaches. Retinal structure images were acquired via Spectral-Domain–Optical Coherence Tomography (SD-OCT). Our results show that the treatment with M9 significantly reduces inflammatory processes in in vitro, ex vivo, and in vivo models of DR. M9 works by inhibiting the proinflammatory responses during DR progression mainly affecting immune cell responses. It also induces an anti-inflammatory response, primarily mediated by microglial cells, leading to the synthesis of Arginase-1 and Hemeoxygenase-1(HO-1). Ultimately, in vivo administration of M9 preserves the retinal integrity from the degeneration associated with DR progression. Our findings demonstrate a specific interaction between both retinal and systemic immune cells in the progression of DR, with a differential response to treatment, mainly driven by microglia in the anti-inflammatory action. In vivo treatment with M9 induces a switch in immune cell phenotypes and functions that contributes to delaying the DR progression, positioning microglial cells as a new and specific therapeutic target in DR.

## 1. Introduction

Diabetes is a major public health problem, affecting 425 million people worldwide, and this number is expected to increase due to the rising prevalence of type 1 diabetes (T1D) and the growing incidence of type 2 diabetes [1]. Despite the different etiologies of type 1 and type 2 diabetes, both are associated with numerous complications that affect the cardiovascular system, kidneys, eyes, and nerves [2].

Diabetic retinopathy (DR) is considered a microvascular disorder that leads to neurodegenerative processes that present late clinical characteristics even after a few years of poorly controlled diabetes. These neurodegenerative processes reflect the nature of the retina’s architecture and its cellular composition [3]. In the case of DR, clinical signs typically manifest sometime after the onset of diabetes, often years later [4]. Over the past decades, inflammation has been recognized as a principal process responsible for various physiological and molecular changes observed in the retinas and vitreous humor of diabetic animals and patients [5]. Increasing evidence points to inflammation as a critical contributor to the development of DR and its regulation can be beneficial in preventing irreversible vascular and neuronal disturbances over time [6,7].

A critical pathological hallmark of DR is the alteration of the blood–retinal barrier (BRB), which is associated with the inflammatory events that accompany the progression of DR [8]. In the case of BRB breakdown, it can be affected by the presence of circulating monocytes, tissue-resident macrophages, or microglia cells, as well as monocyte-derived inflammatory macrophages affecting the retina [9]. In several models of retinopathy, it has been shown that the circulating macrophages migrate to the site of damage, activating microglia cells, triggering the secretion of inflammatory mediators, and modulating the inflammatory response [10]. Emerging evidence indicates that recruited macrophages react differently to neuroinflammation than microglia [11,12].

Retinal microglial cells, specific retinal immune cells with different functional shapes linked to multiple functions, exhibit a rapid response under inflammatory signals, releasing cytokines, chemokines, neurotrophic factors, and neurotransmitters that contribute to the exertion of cytotoxic, cytoprotective, and scavenger actions depending on the tissue context [13]. During retinal neuroinflammation, microglial cells become activated and produce inflammatory mediators in either a pro- or anti-inflammatory response [14,15].

Currently, there is a growing interest in the search for new anti-inflammatory agents, given the key role that inflammation plays in the development of DR [16] and other diseases. In this regard, naturally occurring phthalides are a relatively small group of metabolites produced by various plants and fungi in terrestrial and marine environments. Phthalides present numerous pharmacological effects [17,18], and we have recently shown that the synthetic derivative 3-(2,4-dihydroxyphenyl)phthalide (hereinafter, M9) has strong anti-inflammatory activity [19].

The involvement of inflammation and the immune response in DR has prompted a re-consideration of the triggers for DR development. Novel approaches are now focusing on the role of inflammation and immune system modulation in the pathogenesis of DR. The present work demonstrates the anti-inflammatory effects of a synthetic compound with bioactivity on the progression of DR not only by inhibiting the proinflammatory response but also by inducing the anti-inflammatory response, which is primarily mediated by microglia in the retina.

## 2. Results

### 2.1. Deciphering the Anti-Inflammatory Effects of M9 in LPS-Stimulated Immune Cells

First, we evaluated the cytotoxicity of the 3-arylphthalide M9 on Bv.2 microglial (Figure 1A) and Raw264.7 macrophage cells (Figure 1B) using a dose–response stimulation. Based on these results, we used the 10 µM concentration for subsequent experiments in both cell lines. To determine the potential anti-inflammatory effects of M9, Bv.2 microglial and Raw264.7 macrophage cells were cultured for 24 h in the presence or absence of different M9 doses and stimulated with LPS (200 ng/mL) [20,21] in a co-treatment regimen. As shown in Figure 1C (microglial cells) and 1D (macrophage cells), the increase in nitrite secretion in the culture media induced with LPS was significantly inhibited via treatment with M9 in a dose-dependent manner. The 10 µM concentration of M9 showed the most effective effects on nitrites secretion in LPS-stimulus.

Consistently, a significant reduction in *Nos2* mRNA expression and iNOS protein levels was observed in LPS-stimulated and M9-treated Bv.2 microglial cells (Figure 2A,C) or Raw264.7 macrophage cells (Figure 2B,D), compared to the untreated ones. As depicted in Figure 2E,F, the significant increases in *Il1b* and *Il6* transcripts induced via LPS stimulation were significantly reduced in both immune cell lines after M9-treatment. However, the elevated *Tnfa* mRNA levels after LPS-stimulation only decreased in Bv.2 cells treated with M9, while Raw264.7 cells were not affected at all.

### 2.2. M9 Exerts Anti-Inflammatory Effects via Inhibition of Inflammasome Complex in Immune Cells

We studied the inflammasome complex, as a mechanism involved in the anti-inflammatory effects of M9. As can be seen in Figure 3A,B, the mRNA levels of *Nlrp3* showed, as expected, a significant increase after LPS stimulation, but they were markedly reduced, in both immune cells, after treatment with M9. Additionally, we assessed the effect of this compound on the inflammasome by examining the significant increase in IL1β pro-form and its cleaved form induced via LPS stimulation. The IL1β-activated fragments present a non-canonical molecular weight of 28 KDa; however, the processed 28 kDa form of IL-1β is capable of activating further downstream proinflammatory cytokines as recent articles have shown [22]. In both immune cells, in Bv.2 cells (Figure 3C) and Raw264.7 cells (Figure 3D), the treatment with LPS plus M9 reduced significantly the IL1β pro-form cleavage.

### 2.3. HO-1 Induction Involves the Arginase-1 Upregulation due to M9 in Bv.2 Microglial Cells

Previous studies have shown that HO-1 induction exerts protective effects in cells stimulated by proinflammatory cytokines through the down-regulation of inflammatory mediators and promoting the anti-inflammatory responses mediated by arginase-1 [21,23,24,25]. As Figure 4A,B show, M9 increased HO-1 levels in microglial and macrophage cells either alone or in the presence of LPS. Figure 4C,E show that M9 counteracted the effect of LPS on *Arg1* (gene encoding arginase-1) mRNA levels as well as protein levels. However, in M9-treated Raw264.7 cells, the expected effects on the M2-induced response were only significantly relevant at *Arg1* mRNA levels, but not in arginase-1 protein levels (Figure 4D,F).

### 2.4. The M9 Immunomodulatory Effects Alter/Affect Kinase Stress Signaling Pathways in the Microglial Cell Line Bv.2

To analyze if stress kinase pathway modulation is involved in the anti-inflammatory effect exerted by M9, we examined MAPKs and NFκB-mediated signaling pathways. Treatment of Bv.2 microglial cells with LPS rapidly activated MAPKs by inducing the phosphorylation of JNK and P38α-MAPK, with a maximal effect being elicited after 30 min and maintained until 90 min (Figure 5A,B).

Co-treatment with M9 prevented the nuclear translocation of P65-NF-κB (Figure 5C). Indeed, the LPS plus M9 condition promotes a significant reduction in the levels of phosphorylation of JNK and P38α-MAPK compared to treatment with LPS alone. M9 is also able to induce specific phosphorylation in P38α-MAPK at 60 min and 90 min, which is similar to what was previously described [21], suggesting the alternative pathway of P38α-MAPK activation [26] with anti-inflammatory effects [27,28] in microglial cells, as Figure 4 shows. As Appendix A shows, M9 presents a weaker effect on the JNK and P38α-MAPK pathway in the macrophage cell line. And the P38α-MAPK phosphorylation under a single M9 stimulation was not detected in macrophage cells.

### 2.5. Macrophage–Microglia Interaction under Proinflammatory Environment and the M9 Effects

Associated with retinal early inflammation, the BRB integrity becomes compromised [29], favoring infiltration of systemic immune cells and interaction with immune retinal cells (microglia) by secreting inflammatory mediators. In Figure 6A, we detected increased macrophage markers *Mcp1* and *Cd68* mRNA expression in the retina of BB rats compared to WT rats at 7 weeks. To decipher the potential counteracting effects of immune cells interaction in the retina, the culture medium from Raw264.7 or Bv.2 cells previously stimulated with LPS with or without M9 co-treatment, as it has been described in the material and methods section, was used as a conditioned medium. Each conditioned medium includes the inflammatory mediators secreted by macrophages or microglia, respectively. Bv.2 microglial cells were stimulated with conditioned media from the Raw264.7 cells and vice versa. As shown in Figure 6B, the LPS-conditioned media promotes an increase in nitrite secretion by Bv.2 microglial cells, but it was significantly reduced in the presence of M9. Similarly, *Nos2* mRNA expression in Bv.2 microglial cells treated with LPS-plus-M9-conditioned media was only slightly induced when compared to LPS-conditioned media.

As an anti-inflammatory marker, *Arg1* expression levels were not reverted via M9 treatment under conditioned media (Figure 6C), but *Hmox1* expression levels were increased. In this indirect proinflammatory stimulation, the gene expression of proinflammatory cytokines such as *Il1b* and *Tnfa* was significantly less induced with LPS plus M9 treatment than with LPS alone (Figure 6D). However, the mRNA *Il6* levels, although it tends to decrease, remained without significant differences between both conditions. The LPS-conditioned media from Bv.2 cells do not induce any inflammatory or anti-inflammatory response on Raw264.7 cells (Appendix A).

Analyzing the immune interaction in a physiological system, 7-week-old WT rat retinas cultured in a conditioned medium from macrophages stimulated with LPS were capable of generating a proinflammatory response. The treatment with M9 was capable of partially reversing some proinflammatory markers such as the expression of *Nos2, Il6, Il1b, Nlrp3*, and *Tnfa* (Appendix A) but did not modulate *Arg1* expression (Appendix A). As expected, the conditioned medium from microglia stimulated with LPS was capable of inducing a proinflammatory response in 7-week-old WT rat retinas which was reverted in the presence of M9, with the induction of *Arg-1* mRNA expression (Appendix A).

### 2.6. Retinal Explants from BB Rats Treated with M9 Reduce the Inflammatory Response and Potentially the M2 Response Mediated by Microglial Cells

Neuroinflammation is an early event in DR and precedes neurodegeneration processes associated with DR in the late stages. In an ex vivo approach, retinal explants from 7-week-old BB rats were treated for 24 h with M9 (20 µM); this dose was judged optimal after the previous dose–response study.

The analysis of proinflammatory cytokines revealed a classical reduction in mRNA levels of *Tnfa, Il1b*, and *Il6*, and a significant reduction in mRNA levels of *Nlrp3* was observed, indicating an inhibition of the inflammatory response (Figure 7A). The M9 treatment reduces the iNOS levels detected in retinal explants from BB rats. Furthermore, the protein levels of arginase-1 and HO-1 were significantly increased when retinal explants from BB rats were cultured in the presence of M9 (Figure 7B). GFAP immunostaining, as a marker of reactive gliosis associated with inflammatory events during DR, was highly expressed in the retinal explants from BB rats at 7 weeks old. However, M9 treatment induced a significant reduction in the GFAP immunoreactive signal (Figure 7C).

### 2.7. M9 Reduces the Inflammatory Phenomena and Modulates the Microglia Polarity towards an Anti-Inflammatory Phenotype during DR without Modifying the Metabolic Mechanisms of Diabetes

To test if M9 could regulate in vivo the inflammatory process in DR, diabetic BB rats at 7 weeks old were treated via intraperitoneal injection with M9 (600 µg/kg/day) or vehicle for two weeks and 3 times per week. No differences in blood glucose levels were found between diabetic rats treated with M9 or vehicle at each time point, and the values are confined to the normoglycemia ratio. Early inflammatory events are detected before blood glucose levels increase in BB rats, as seen in the data in Figure 7. The difference in body weight detected was associated with the normal weight gain in 2 weeks of treatment (Table 1).

Nevertheless, by analyzing the longitudinal changes that occur during retinal degeneration from the OCT images of the BB rats treated with M9 or vehicle for 15 days, we detected a preventive effect of M9 on retinal layer damage in DR progression (from 7- to 9-weeks old). DR progression contributes to the loss of retinal cytoarchitecture, with a reduction in total and inner nuclear layer (INL) thickness observed from 7-weeks old to 9-weeks old. The M9 treatment prevented the reduction in total retinal thickness at the advanced stage of DR, and the outer nuclear layer (ONL) thickness is recovered after treatment (Figure 8A). As the Appendix A shows via OCT images, the BB rats with M9 treatment reduced the neovascularization and the presence of drusenoid deposits and hyper-reflective drusen cores, compared to the BB rat vehicle treatment, maintaining a similar profile to WT rats. The Appendix A corroborate the retinal thickness quantification via the measurement of the retinal layer in DAPI dyed sections from different treatments.

The structural retinal preservation was related to a strong reduction in proinflammatory cytokine expression (Figure 8B) and a significant induction of M2 or anti-inflammatory response, mainly due to microglial actions. As Figure 8C shows, M9 promoted HO-1 and arginase-1 levels in the retina from BB rats with M9 administration and a reduction in NLRP3 levels.

The M9 administration reduced the classical gliosis detected in DR associated with T1DM progression (Figure 8D) and induced a switch in microglial shape towards ameboid microglia associated with the activated phenotype (Figure 8E). According to the M2 response induction detected in vitro and ex vivo approaches, an increase in arginase-1 immunodetection after M9 administration colocalized with IBA-1 immunopositive microglial cells (Figure 8E). In Figure 8F, the quantification of dual markers indicated that M9 treatment induced an anti-inflammatory phenotype in activated microglial cells.

## 3. Discussion

The inflammatory phenomena that occur during DR typically contribute to the progression of retinal pathology [8]. This phenomenon can be exacerbated by the activation of the retinal innate immune system and, additionally, by the peripheral immune system due to increased permeability of the BRB [11,12]. The modulation of early processes in DR can lead to a better therapeutic response and improved patient outcomes. We analyzed the immunomodulation resulting from the inflammatory response of microglia during DR and its potential as a therapeutic target. The administration of bioactive compounds, such as M9, may represent a new strategy in the treatment of early events, including inflammation and the progression of DR, with a more favorable prognosis. To this end, the anti-inflammatory effect of M9 has been evaluated in immune cell cultures of microglia and macrophages, as well as in ex vivo retinal explants and through in vivo administration in type 1 diabetic BB rats. This compound has been demonstrated to reduce inflammation, to promote an anti-inflammatory response, and to delay DR progression

Treatment of Bv.2 microglial and Raw264.7 macrophage cells with M9 in a LPS co-stimulation condition reduced NO production and inhibited LPS-induced *Nos2* and proinflammatory cytokine mRNA expression in both immune cell lines compared to the LPS-stimulation condition. These results are supported by the reduction in classical proinflammatory cytokine expressions such as *Il1b* and *Il6*, but only in Bv.2 microglial cells was *Tnfa* expression abolished. It is important to note that M9 promoted the upregulation of HO-1, which plays a crucial role in responding to cellular damage, inducing the expression of arginase-1 which was used as a marker of the anti-inflammatory or M2 response [30], preventing NLRP3 inflammasome activation, decreasing IL-1β secretion, and minimizing the inflammatory reaction [31]. In this regard, M9 treatment predominantly induced the M2 response in Bv.2 microglial cells, while Raw264.7 macrophage cells displayed an M2 response although with a weaker effect on arginase-1 synthesis. Inflammasome complex regulation is mediated by different mediators including P65-NF*k*B as a nuclear transcription factor [32]. The proinflammatory signaling pathway involved in these events showed that M9 prevents LPS-mediated nuclear translocation of P65-NF*k*B and decreases *Nlrp3* mRNA expression.

The effects of M9 act within the classical proinflammatory environment induced with LPS, attenuating JNK and MAPK signaling pathways in both microglial and macrophage cells. However, the effects of M9 on the anti-inflammatory response are compromised in macrophages, being unable to decrease the elevated levels of *Tnfa* expression induced with LPS. Recent studies have demonstrated that the modulation of the anti-inflammatory response with HO-1 depends on P38α-MAPK activation [33,34]. Different natural compounds induce P38α-MAPK autophosphorylation upon binding to the allosteric lipid binding site, causing a conformational change that promotes the induction of HO-1 and arginase-1 [21,25]. M9 treatment induces a similar P38α-MAPK phosphorylation in Bv.2 microglial cells, and these results align with the specific M2 induction observed with M9 treatment in Bv.2 microglial cells, as evidenced by the induction of HO-1 and arginase-1. However, the ability of M9 to induce P38α-MAPK phosphorylation in Raw264.7 macrophage cells was not detected, contributing to non-M2 response detection, in the same way that occurs with other bioactive compounds [21]. Therefore, treatment with M9 induces non-canonical P38α-MAPK activation, specifically restricted to the response of microglial cells. These results suggest that microglial and macrophage cells exert an anti-inflammatory response to M9 via different signaling pathways.

However, despite having evidence of the breakdown of the BRB during DR and the interconnected communication of the immune systems involved [12,35], the macrophages could contribute to enhancing the pro- or anti-inflammatory response regulated by microglia and the proinflammatory environment could explain the dramatic progression of DR in some patients [36]. In our studies, the effects mediated by the macrophage-conditioned medium on microglia reveal that the population of microglial cells respond to the inflammatory stimulus of macrophages. However, as was expected, the responses of macrophages to microglial-conditioned medium do not generate any inflammatory or anti-inflammatory response. Inflammation is a consortium of mediators that act coordinately in order to respond to a perceived stimulus [37], so the absence of response in macrophages in cell cultures was explored in a more physiological environment, such as organotypic or retinal explant. The proinflammatory environment that mimics the inflammation present during DR was reproduced in WT retinal explants cultured with macrophage-conditioned media, leading us to conclude that the interaction between macrophages and microglia is not bidirectional. The main inflammatory response in DR is primarily attributed to the activation of microglia and its responsiveness to treatment.

As previous results in retinal explants from BB rats and *db/db* mice have demonstrated, treatment with different natural molecules reduces inflammatory events during DR [21,25,38,39]. DR progression is linked to inflammatory events that precede retinal failure [40]. The BB rats, an animal model for T1DM, are characterized by their ability to mimic human DR progression and are an interesting animal model for studying potential pharmacological treatment options in preclinical studies [41,42]. The analysis of retinal inflammation in diabetes progression has focused on the presence of inflammatory markers in retinal tissue and their immunomodulation [43,44].

This study demonstrates that in both ex vivo (retinal explants) and in vivo BB rat models, M9 treatment reduces proinflammatory cytokine mRNA expression and induces the anti-inflammatory markers of arginase-1 and HO-1, which are active modulators of inflammasome complex activation. The beneficial effects of M9 administration were evident not only at the molecular level, but also in the preservation of the retinal cytoarchitecture and the maintenance of the thickness of the retinal layers in treated animals. These parameters are significantly associated with the maintenance of synaptic connections between the layers of neurons in the retina, thus preserving visual function [45,46]. The inflammatory events in DR concur with the classic retinal cell markers associated with inflammatory events, such as activated microglia (IBA-1) and macroglia (GFAP), showing evidence of these processes in both the in vivo and ex vivo approaches. Reactive gliosis reduced via M9 treatment shows a retraction in cell projections consistent with a lower state of inflammation. Previous evidence supports the use of anti-inflammatory treatments for DR, as they can promote a change in the phenotype and function of microglial cells, which are primarily responsible for the immune response in DR [20]. M9 administration increased the detection of arginase-1 in the retina, primarily colocalized with IBA-1 immunopositive cells. Our data corroborate that the treatment with M9 induces the shift from an M1 to an M2 stage during DR, supporting the inhibition of proinflammatory and the induction of anti-inflammatory responses. The polarity change towards the M2 phenotype is probably modulated by non-canonical phosphorylation of P38α-MAPK in microglial cells that contributes to the induction of HO-1 and arginase-1 [47]. In diabetes and oxidative stress environments, increased expression of arginase-1 leads to the hydrolysis of L-arginine into urea and L-ornithine [48]. Arginase-1 and nitric oxide synthase isoforms (endothelial-eNOS and inducible –iNOS) compete for a common substrate, L-arginine. Elevated levels of arginase-1 in immune cells reduces the bioavailability of L-arginine, which in turn reduces iNOS expression and function, resulting in M1-phenotype inhibition [49]. In the same way, the increased production of polyamines mediated by elevated arginase-1 levels promotes the polarization of microglia from the M1 toward the M2 phenotype [50].

We conclude that microglial cells are mainly responsible for the anti-inflammatory response during DR, helping to counteract the proinflammatory environment. M9 administration exerted significant and beneficial effects by inhibiting the proinflammatory response, modulating P38α-MAPK phosphorylation, promoting M2-response induction, protecting the retinal structure from degenerative processes, and aiding to eliciting an M2 phenotype in microglial cells.

Modulating the immune response directly through the administration of new bioactive molecules offers a promising new therapeutic target. The change in the polarization of microglia could become a strategy to combat the progression of DR, regardless of the disease stage. The bioactive compound M9 holds substantial potential due to its anti-inflammatory properties and its capability to promote the resolution of inflammation during DR.

## 4. Materials and Methods

### 4.1. Reagents

Please see the Appendix A for full experimental details.

### 4.2. Antibodies

Please see the Appendix A for full details.

### 4.3. Synthesis of the 3-arylphthalide M9

The compound 3-(2,4-dihydroxyphenyl)phthalide (M9) was synthesized as previously described [19].

### 4.4. Cell Culture

The mouse microglial cell line Bv.2 was purchased from ACCEGEN Biotechnology (ACCEGEN Biotechnology, Fairfield, USA). The mouse macrophage cell line Raw264.7 was provided by Dr. A.M. Valverde (IIBm “Alberto Sols” UAM-CSIC-Madrid, Spain). Cells were used for experiments in passages 12–22. Please see the Appendix A for full experimental details.

### 4.5. Analysis of the Cellular Viability by Crystal Violet Staining

The cells were cultured in serum-free media and stimulated in the presence or absence of M9 (0.1, 0.5, 1, 10, and 25 µM) for 24 h to assess viability using crystal violet staining [25]. Please see the Appendix A for full experimental details.

### 4.6. Analysis of Nitrite (NO_2_^−^)

To determine the nitrites production through the Griess test [51], the cells were stimulated with LPS (200 ng/mL), which mimics the diabetic proinflammatory environment [20], in the presence or absence of M9 for 24 h. Please see the Appendix A for full experimental details. In additional experiments, Bv.2 and Raw264.7 cells were cultured in a co-treatment regimen with M9 (10 µM) and LPS (200 ng/mL) for 24 h.

### 4.7. Conditioned Medium from Immune Cells

To obtain conditioned media, immune cells were stimulated with LPS for 8 h in the presence or absence of M9. Media were then replaced with fresh media, with the absence of LPS, and incubated for an additional 16 h. In this way, the conditioned media contained the inflammatory mediators induced via LPS stimulation but not LPS that could interact with the cell or retinal explant host. These conditioned media were carefully harvested, filtered using sterilized 0.2 µM filters, and stored at −80 °C for further experiments involving Bv.2 cells or retinal explants.

### 4.8. Animals

All animal procedures were performed with the approval of the Committee for the Ethical Use and Care of Experimental Animals (University of Cadiz, Spain), Protocol number 005_abr20_PI2–ITI-012-2019. Animal experimentation conducted in this study followed the recommendations of the Federation of European Laboratory Animal Science Associations (FELASA) on health monitoring in accordance with the regulations of the Association for Research in Vision and Ophthalmology (ARVO).

Bio-Breeding (BB) and Wistar rats were maintained under conventional conditions in an environment-controlled room (20–21 °C and 12 h light–dark cycle) with water and standard laboratory rat chow available ad libitum. Blood samples from the tail vein were used in BB rats for weekly random glucose measurements using an automatic glucose monitor (Freestyle Optium Neo, Abbott, Madrid, Spain). Diabetes onset was defined by glucose levels above 270 mg/dL (14.98 mmol/L).

### 4.9. Retinal Explants

*Ex vivo* assays were performed with retinas from 7-week-old male or female BB rats. The rats were euthanized via an overdose of anesthesia, and the eyes were enucleated. The lens, anterior segment, vitreous body, retinal pigment epithelium, and sclera were removed. The retinas were immediately cultured in R16 media (provided by Dr. P.A. Ekstrom, Lund University, Sweden) with no additional serum on cell culture inserts with a pore size of 0.4 μm. Retinas were cultured with or without M9 at 20 µM as indicated in the figure legends.

### 4.10. Intraperitoneal Administration of M9

A total of 10 female or male BB rats 7-weeks old were randomly divided into the M9 group and the control group. Rats in the M9 group were treated via intraperitoneal administration three days per week for two weeks (days 1–15) at the dosage of 600 µg/kg/day. The control group received an equal volume of vehicle on the same days. The dose and time of treatment were previously determined. Blood glucose levels and body weight on days 0, 7, and 15 were determined, and the eyeball or retinal tissue was processed for immunofluorescence, protein, or RNA extraction. Please see the Appendix A for full experimental details.

### 4.11. Spectral-Domain Optical Coherence Tomography (SD-OCT)

OCT images were obtained using a Micron IV rodent imaging system (Phoenix Research Labs, Pleasanton, CA, USA) as described previously [52]. The thickness of both eyes (containing average raw data from 624 A-scans from each eye after removing the first and last 200 scans) was averaged per animal for statistical analysis. The overall average retinal layer thickness was presented as the mean ± standard error. The number of rats used in each measurement is described in Appendix A. Please see the Appendix A for full experimental details.

### 4.12. Immunofluorescence

Bv.2 microglial cells were seeded on coverslips 24 h before LPS stimulation and/or M9 treatment in serum-free media. Cytosolic or nuclear P65 NFkB immunolocalization was determined as previously has been reported [27]. For retina immunofluorescence analysis, eye cryosections and whole retina explants were processed in a similar way as described previously [27] for immunodetection of GFAP, IBA-1, and Arginase-1. The staining was observed and recorded with an inverted laser confocal microscope Axiovert (ZEISS, Jena, Germany). Please see the Appendix A for full experimental details.

### 4.13. Western Blot

Immunoblotting against several antibodies (Appendix A) was performed. Please see the Appendix A for full experimental details.

### 4.14. Quantitative Real-Time Polymerase Chain Reaction (qRT-PCR) Analysis

Mouse and rat Taqman probes for transcripts *Tnfa*, *Il6*, *Il1b*, *Il10*, *Nlrp3*, *Nos2*, *Arg1, Hmox1*, and *Gapdh* (Appendix A) were purchased from Applied Biosystems (Waltham, MA, USA). Please see the Appendix A for full experimental details.

### 4.15. Statistical Analysis

Western blot quantification was performed using the ImageJ program. Values in all graphs are presented as means ± SEM. Statistical tests were performed using GraphPad Prism7.0a Software (Boston, MA, USA). Data were analyzed using one-way ANOVA followed by Bonferroni test or Student paired t-test when comparisons were between any two groups. Differences were considered significant at *p* < 0.05.

## Figures and Tables

**Figure 1 ijms-25-08440-f001:**
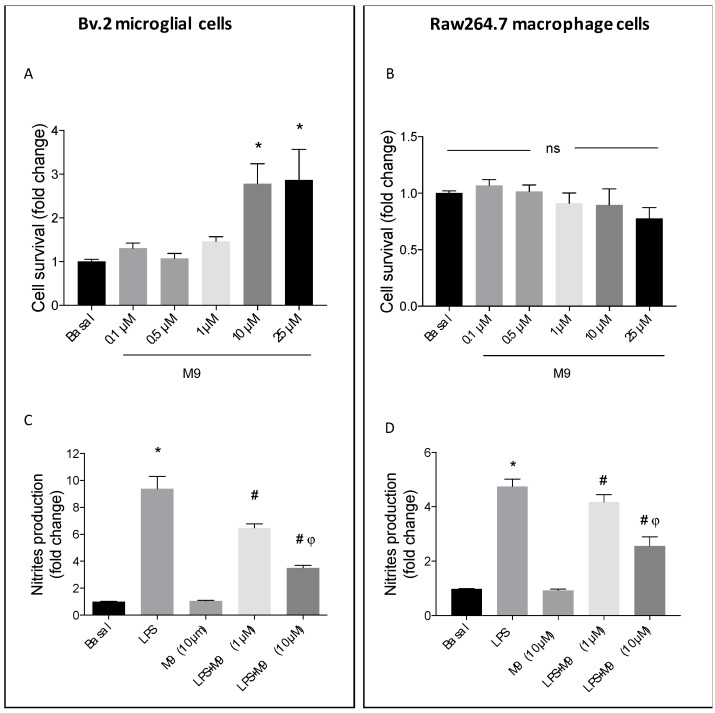
M9 effects on cellular viability and nitrites production in LPS-stimulated microglial and macrophage cells. Viability was determined using crystal violet staining in Bv.2 microglial cells (**A**) and Raw264.7 macrophage cells (**B**). Cells were treated for 24 h with different concentrations of M9 (0.1–25 μM). Nitrites accumulation was analyzed and compared to the basal levels in Bv.2 microglial cells (**C**) and Raw264.7 macrophage cells (**D**). Cell cultures were treated with LPS (200 ng/mL), M9 (10 μM), or LPS plus M9 (1 and 10 μM) for 24 h. The results are presented as mean ± S.E.M. The fold change relative to the basal condition is shown. (*n* = 4 independent experiments) * *p* ≤ 0.05 vs. basal, ^#^ *p* ≤ 0.05 vs. LPS, and ^φ^ *p* ≤ 0.05 vs. LPS + M9 (1 μM) treatment (one-way ANOVA followed by Bonferroni *t*-test). ns (no significant differences).

**Figure 2 ijms-25-08440-f002:**
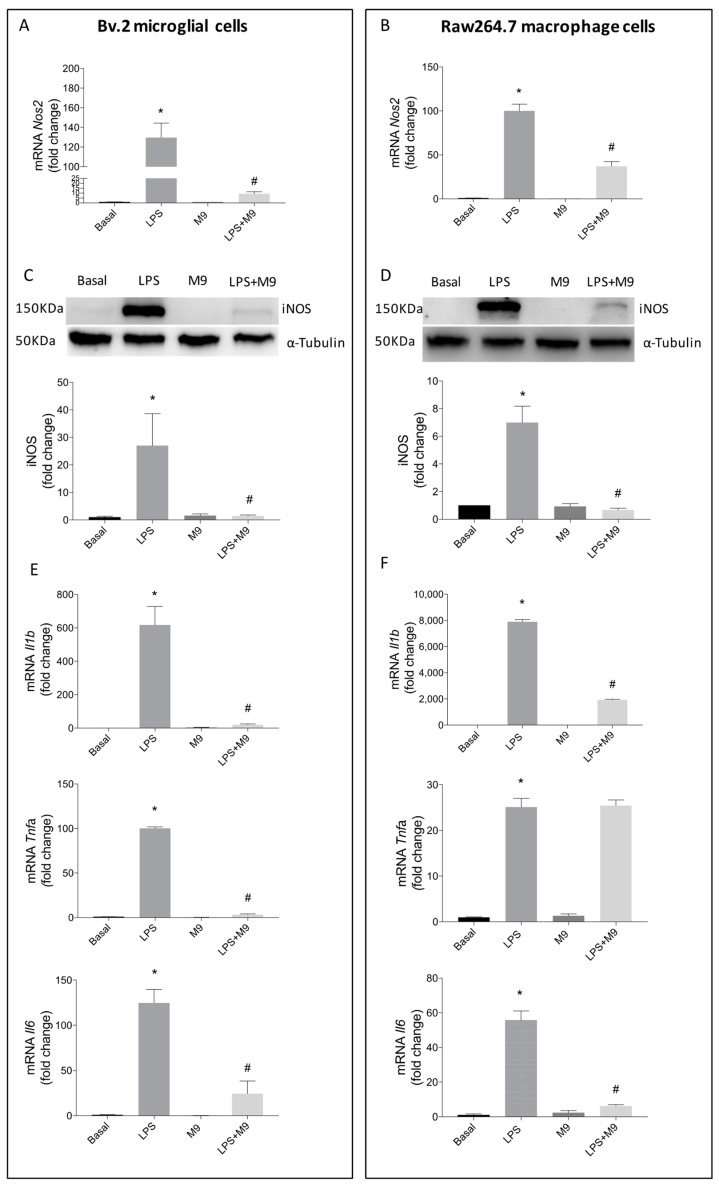
Protective effects of M9 against LPS stimulation of proinflammatory mediators in microglial and macrophage cells. *Nos2* mRNA values were determined using qRT-PCR in Bv.2 microglial cells (**A**) and Raw264.7 macrophage cells (**B**) after treatment with LPS (200 ng/mL), M9 (10 µM), or LPS plus M9 for 24 h. iNOS protein levels were analyzed using Western blot in protein extracts from above Bv.2 microglial cells (**C**) or Raw264.7 macrophage cells (**D**) treated with LPS, M9, or LPS + M9. α-tubulin was used as loading control. *Il1b*, *Il6*, and *Tnfa* mRNA levels were determined using qRT-PCR in Bv.2 microglial cells (**E**) and Raw264.7 macrophage cells (**F**). Data were normalized to *Gapdh* gene expression. The results are presented as means ± S.E.M (*n* = 5 independent experiments). Fold changes are calculated relative to the basal value. * *p* ≤ 0.05 vs. basal treatment, ^#^ *p* ≤ 0.05 vs. LPS treatment (one-way ANOVA followed by Bonferroni *t*-test).

**Figure 3 ijms-25-08440-f003:**
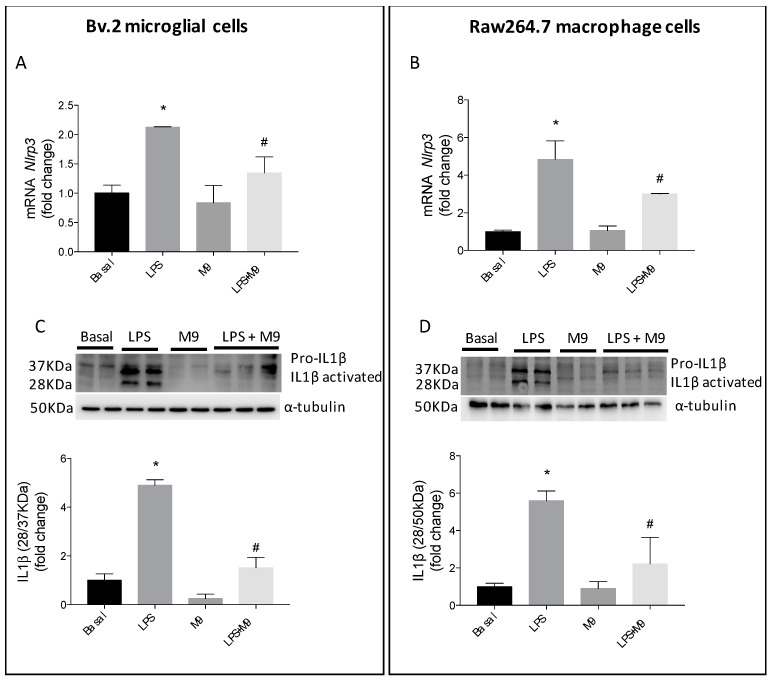
Protective effects of M9 against LPS-mediated activation of the inflammasome in microglia and macrophage cells. Bv.2 microglial cells and Raw264.7 macrophage cells were treated for 24 h with LPS (200 ng/mL) or LPS plus M9 (10 µM). (**A**) Bv.2 microglial cells and (**B**) Raw264.7 macrophage cells; *Nlrp3* mRNA levels were determined using qRT-PCR. Data were normalized to *Gapdh* gene expression. Protein extracts from Bv.2 microglia cells (**C**) and Raw264.7 macrophage cells (**D**) were analyzed using Western blot with antibody against IL1β. α-Tubulin was used as a loading control. The results are presented as mean ± S.E.M. The fold change relative to the basal condition is shown. * *p* ≤ 0.05 vs. basal treatment, ^#^ *p* ≤ 0.05 vs. LPS treatment (one-way ANOVA followed by Bonferroni *t*-test).

**Figure 4 ijms-25-08440-f004:**
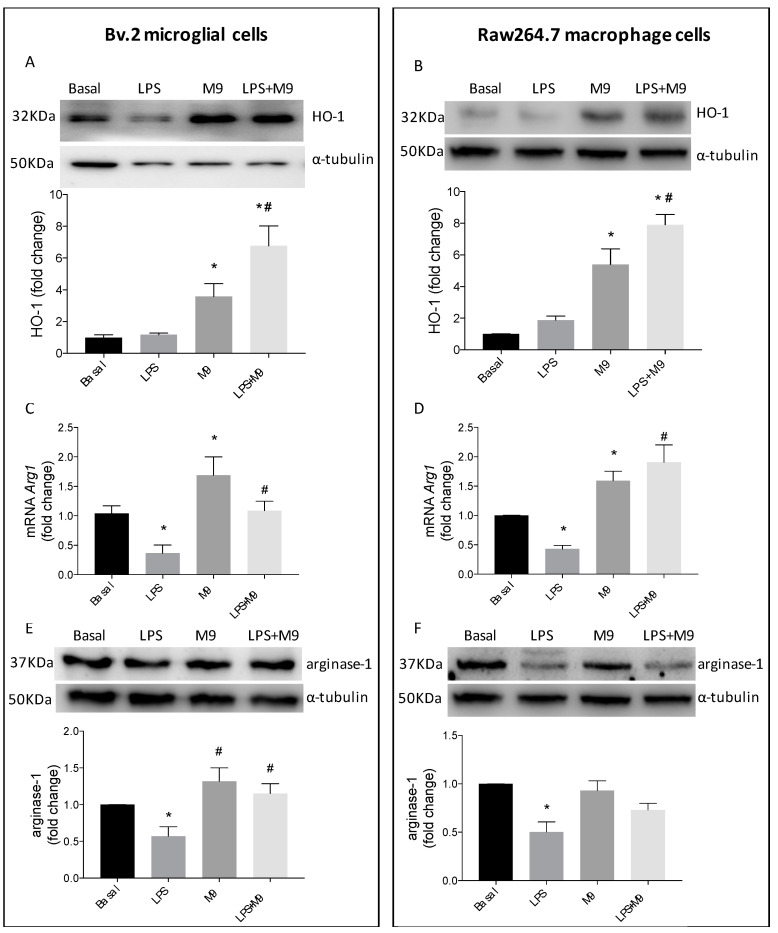
The anti-inflammatory response is mediated by HO-1 and arginase-1 in Bv.2 microglial and Raw264.7 macrophage cells. Bv.2 microglial cells and Raw264.7 macrophage cells were treated for 24 h with LPS (200 ng/mL) or LPS plus M9 (10 µM). Protein extracts from Bv.2 microglial cells (**A**,**E**) and Raw264.7 macrophage cells (**B**,**F**) were analyzed using Western blot with antibodies against HO-1 and arginase-1. α-Tubulin was used as a loading control. *Arg1* mRNA levels in Bv.2 microglial cells (**C**) and Raw264.7 macrophage cells (**D**) were determined using qRT-PCR. Data were normalized to *Gapdh* gene expression The results are presented as mean ± S.E.M. The fold change relative to the basal condition is shown. * *p* ≤ 0.05 vs. basal treatment, ^#^ *p* ≤ 0.05 vs. LPS treatment (one-way ANOVA followed by Bonferroni *t*-test).

**Figure 5 ijms-25-08440-f005:**
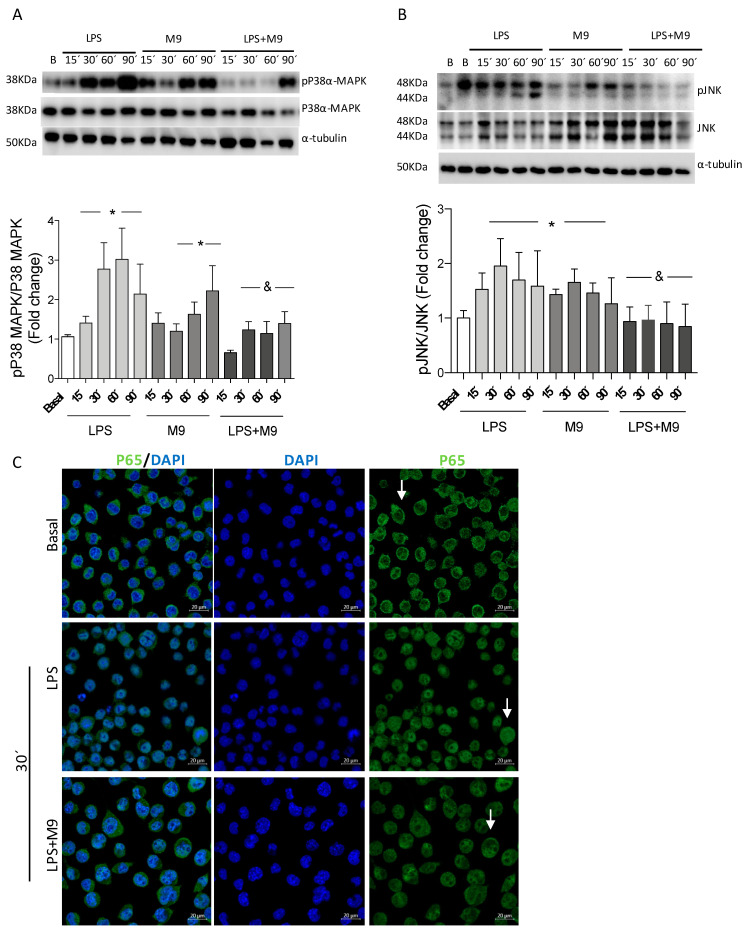
M9 inhibited the activation of NFkB-mediated signaling with P38α-MAPK phosphorylation in LPS-stimulated microglial cells. Bv.2 microglial cells were treated for 24 h with LPS (200 ng/mL) or LPS plus M9 (10 µM) for the time course indicated. (**A**) Protein extracts were analyzed using Western blot with antibodies against phosphorylated(p)-P38α MAPK, total P38α-MAPK, (**B**) phosphorylated (p)-JNK, and total JNK. α-Tubulin was used as a loading control. The results are presented as mean ± S.E.M. The ratios between the indicated proteins and the fold changes relative to the basal values are shown. * *p* ≤ 0.05 vs. basal treatment, ^&^ *p* ≤ 0.05 vs. LPS treatment (one-way ANOVA followed by Bonferroni *t*-test). (**C**) Confocal immunofluorescence assessment of the nuclear translocation of P65-NFkB (green channel). Nuclear regions were determined by counterstaining nuclear DNA with DAPI (blue channel). White arrows indicate the P65-NFkB nuclear or cytoplasmic localization.

**Figure 6 ijms-25-08440-f006:**
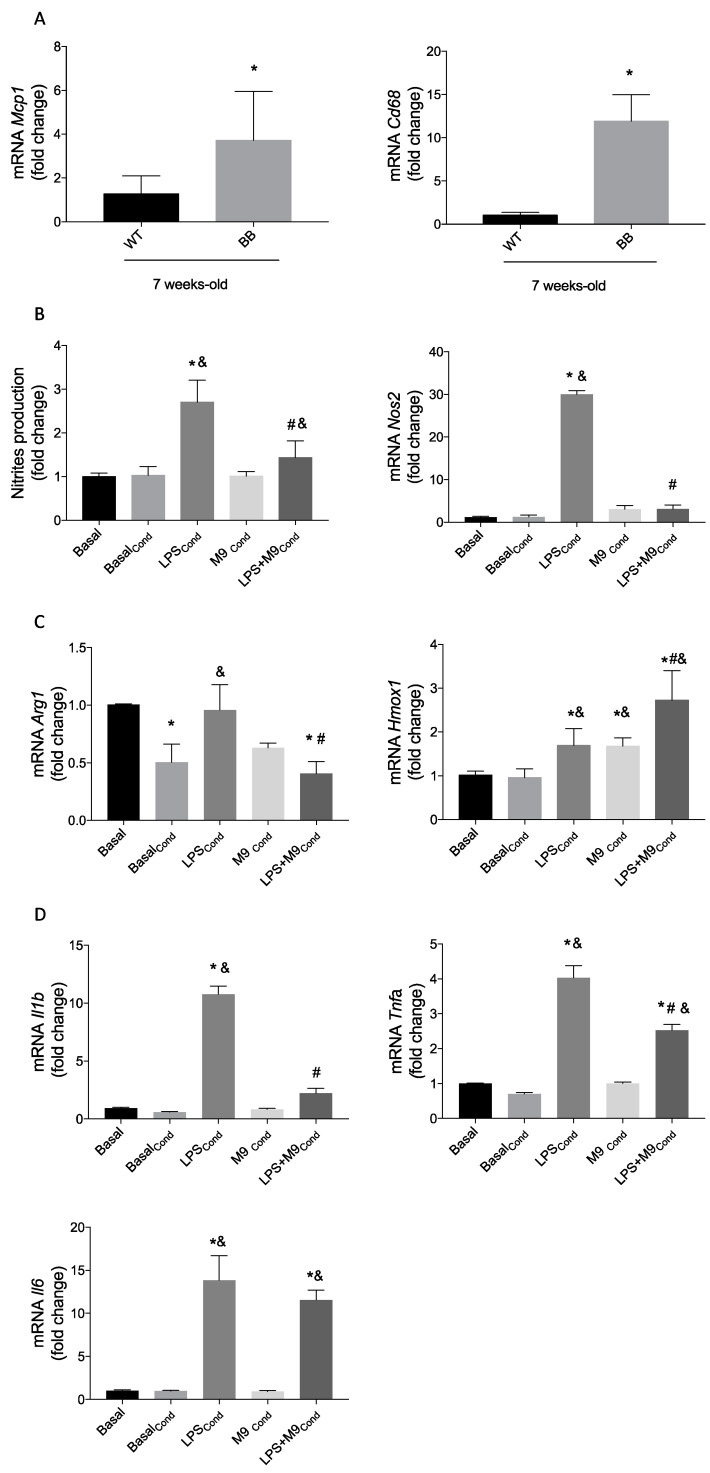
Macrophage-conditioned medium stimulated with LPS or LPS + M9 induces the inflammatory response of the microglia. Macrophage markers in the retina from 7-week-old BB and WT rats. (**A**) *Mcp1* and *Cd68* mRNA values were determined using qRT-PCR. (*n* = 5 retina per condition). Bv.2 microglial cells were treated for 24 h with conditioned medium from Raw264.7 cells cultured previously with LPS (200 ng/mL) or LPS plus M9 (10 µM) for 24 h. (**B**) Nitrite accumulation was analyzed and related to the basal levels, and *Nos2* mRNA values were determined using qRT-PCR. Data were normalized to *Gapdh* gene expression. (**C**) Anti-inflammatory mediators *Arg1* and *Hmox1 mRNA* values were analyzed using qRT-PCR. (**D**) Proinflammatory cytokines *Il1b*, *Tnfa*, and *Il6* mRNA values were determined using qRT-PCR. Data were normalized to *Gapdh* gene expression. The results are presented as mean ± S.E.M (*n* = 4 independent experiments). Fold changes are calculated relative to the basal value. * *p* ≤ 0.05 vs. basal treatment, ^#^ *p* ≤ 0.05 vs. LPS_Cond_ treatment, ^&^ *p* ≤ 0.05 vs. basal_Cond_ treatment (one-way ANOVA followed by Bonferroni *t*-test; or *t*-test).

**Figure 7 ijms-25-08440-f007:**
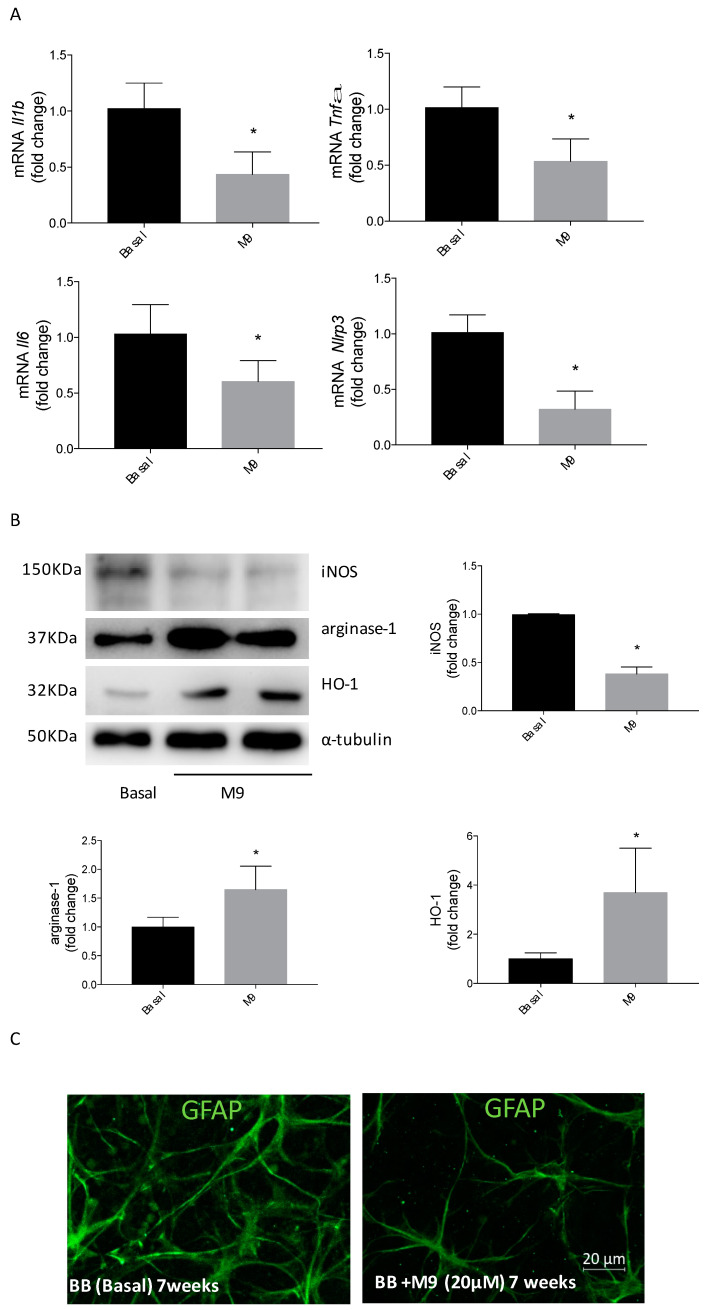
M9 treatment in retinal explants from BB rats decreased inflammatory events and induced anti-inflammatory response. Retinal explants from 7-week-old BB rats were treated for 24 h with M9 (20 μM) or vehicle. (**A**) *Il1b*, *Tnfa*, *Il6*, and *Nlpr3* mRNA values were determined using qRT-PCR. Data were normalized to *Gapdh* gene expression. (**B**) Protein extracts were analyzed using Western blot with antibodies against iNOS, arginase-1, or HO-1. α-Tubulin was used as a loading control. The results are presented as mean ± S.E.M (*n* = 5 retina per condition). The fold change relative to the basal condition is shown. * *p* ≤ 0.05 vs. BB retinal explant basal condition value (*t-*test). (**C**) Retinal explants from 7-week-old BB rats were treated for 24 h with M9 (20 μM) (right panel) or with vehicle (left panel). Immunostaining for GFAP (green) was carried out in whole retinas. Representative images are shown (*n* = 5 retinas per condition).

**Figure 8 ijms-25-08440-f008:**
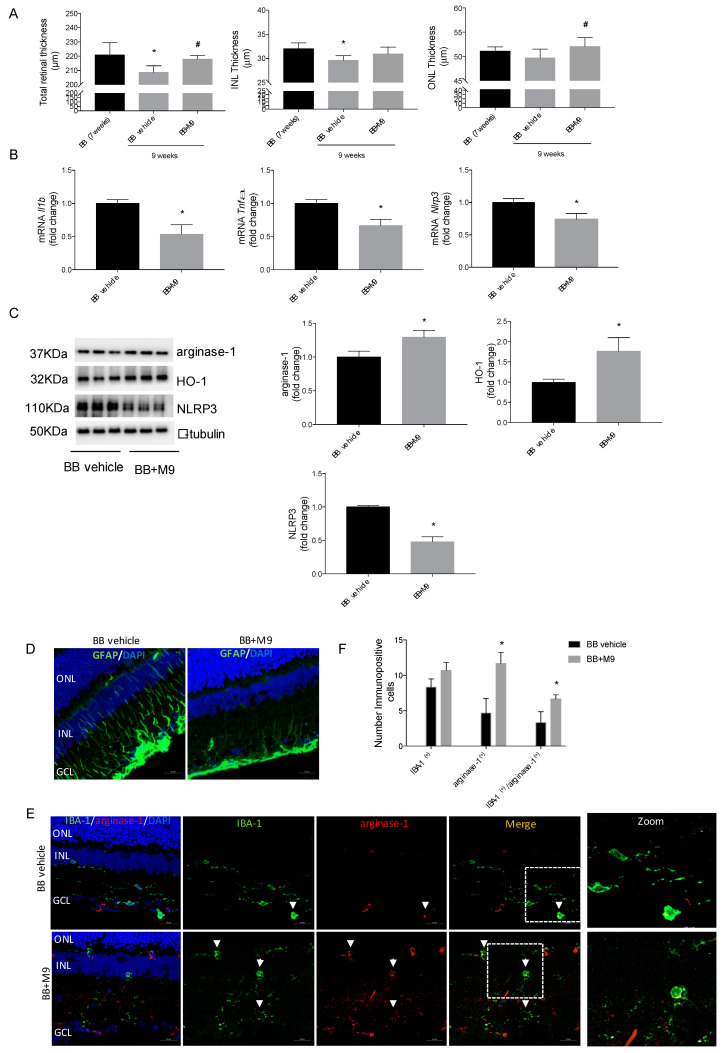
M9 treatment reduces in vivo DR progression in BB rats. BB rats were treated with M9 (600 µg/kg/day) via i.p. for 15 days. (**A**) Total, INL, and ONL retinal thickness measured on SD-OCT. The results are presented as mean ± S.E.M. * *p* ≤ 0.05 vs. BB rat 7-weeks old, ^#^ *p* ≤ 0.05 vs. BB rat vehicle value. (**B**) *Il1b*, *Tnfa*, and *Nlpr3* mRNA values were determined using qRT-PCR. Data were normalized to *Gapdh* gene expression. Fold changes are calculated relative to the basal value. * *p* ≤ 0.05 vs. BB rat vehicle condition value (*t*-test). (**C**) Protein extracts were analyzed using Western blot with antibodies against arginase-1, HO-1, or NLRP3. α-Tubulin was used as a loading control. (**D**) GFAP immunostaining (green) in retinal sections counterstained with DAPI (blue). (**E**) Immunostaining and (**F**) quantification of arginase-1 (red) (arginase-1^+^) and IBA-1 (IBA-1^+^) (green) positive cells in retinal sections counterstained with DAPI (blue). Data are expressed as mean ± SD (*n* = 6 retinas per condition). * *p*-value < 0.05, Student’s *t*-test between IBA-1^+^ and arginase-1^+^ cell subtypes. Scale = 20 μm. Dashed boxes indicate the zoom area showed. White arrows indicate the immune colocalization for arginase-1^+^ and IBA-1^+^ cells (yellow). ONL (outer nuclear layer), INL (inner nuclear layer), and GCL (ganglion cell layer).

**Table 1 ijms-25-08440-t001:** Weight and glycemia in BB rats treated with vehicle or M9 via intraperitoneal injection.

Administration Days.	Weight (g)		Glycemia (mg/dL)	
	Vehicle	M9	Vehicle	M9
0	155.33 ± 31.77	146.80 ± 15.36	111.67 ± 13.42 *#	97.40 ± 8.17
1	160.00 ± 34.69	149.80 ± 16.52	106.67 ± 3.21	107.60 ± 10.47 *#
3	168.67 ± 38.47	156.20 ± 20.10	98.33 ± 7.09	99.20 ± 3.89
6	174.66 ± 42.77	164.20 ± 21.62	90.67 ± 11.84	96.60 ± 4.21
8	177.67 ± 44.45	167.40 ± 24.06	100.00 ± 6	98.20 ± 7.59
10	185.66 ± 46.71	177.00 ± 27.09	92.00 ± 5.29	97.00 ± 6.28
13	195.33 ± 50.14	183.80 ± 28.46	102.67 ± 0.577	98.20 ± 6.76
15	198.33 ± 49.32	186.4 ± 30.82	86.00 ± 6.24	89.00 ± 4.52

* *p* ≤ 0.05 vs. day 15 Vehicle group and # *p* ≤ 0.05 vs. day 15 M9 group.

## Data Availability

The original contributions presented in the study are included in the article/Appendix A, further inquiries can be directed to the corresponding author/s.

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
