# Peer review of "Arylphthalide Delays Diabetic Retinopathy via Immunomodulating the Early Inflammatory Response in an Animal Model of Type 1 Diabetes Mellitus"

_ijms, 2024, doi:10.3390/ijms25158440_

Round 1

Reviewer 1 Report

Comments and Suggestions for Authors

Generally, an interesting drug to target diabetic retinopathy is introduced. Several issues could be addressed to improve the current manuscript.

1. Cell survival increases by M9 treatment. Does this have the clear mechanisms? Data' conditions (Figure 1C) with this effect (Figure 1A) might be affected. The Figure 1C's incubation time point should be shorten (24 h to 6 to 12 h).

2. Under the LPS incubating condition, effects of M9 are dramatic. However, aspects of pathologic diabetic/ischemic conditions in vitro have not been clearly examined. High glucose and/or hypoxia (at least, pseudohypoxia, if experimentally difficult)-stress conditions are needed in this manuscript as the current title as well as the topic is directly about diabetic retinopathy.

3. In Figure 4A' band processing is not convincing (M9's band is half visible; basal band's HO-1 and tubulin are not well aligned). It should be replaced with clear bands.

4. Figure 5A and 5B's alpha-tubulin should not be omitted. Thanks for providing them like the authors did to the other figures' style.

5. BB rats' glucose levels need clear verifications. Is it a clear diabetic condition? Furthermore, M9's glucose levels seem relatively lower. Discussion at least is needed in-depth. 

6. There is no representative image for histology. Total retinal thickness along with inner and outer retinal thickness should be presented with images.

7. How this drug reaches to the retina in vivo should be explained. How this drug absorbs or goes into cells in vitro and in vivo should be explained.

Reviewer 2 Report

Comments and Suggestions for Authors

This contribution deals with the implication of inflammation in the development of retinopathy diabetes (DR). The evidence reported points to inflammation as a critical contributor to the development of DR. Thus, inhibition of the inflammatory process could prevent irreversible vascular and neuronal disturbances. Novel approaches are now focusing on the role of inflammation and immune system modulation in the pathogenesis of DR. Taking advantage that some naturally occurring phthalides found in various plants and fungi display beneficial pharmacological effects,  this group has recently published a paper (ref. 19) proving that the synthetic phthalide derivate 3-(2,4-dihydroxyphenyl)phthalide (called M9 throughout the manuscript) has strong anti-inflammatory activity. The  present work emphasizes the action of M9 and demonstrated that its beneficial effects on DR are exerted by a double mechanism.

Concerning the manuscript details, methods are correct. Two different types of cells, Bv.2 microglial and Raw264.7 macrophage cells. Concerning results, the quantification of  markers at the mRNA and protein level is exhaustive and all data indicated that M9 treatment induced an anti-inflammatory phenotype in activated microglial cells. The use as LPS (200ng/mL) as positive pro-inflammatory agent is also appropriate. According to the length of the manuscript, the section of supp. Material is appropriate, although the separation of some methods in the regular paper and the supp. Material makes sometimes difficult following details.

In sum, I find the contribution exhaustive, consistent and well written.  I have just one minor suggestion. As far as I know, compared to other studies concerning the inflammatory factors, Arginase-1 is not used as a typical marker of the process. The mechanism of the anti-inflammatory response mediated by Arginase-1 and the possible relation with nitric oxide and the corresponding synthases would be briefly discussed

Round 2

Reviewer 1 Report

Comments and Suggestions for Authors

Questions and comments are well-responded.

Author Response

Thank you very much for your comments and suggestions